# UVB-Pretreatment-Enhanced Cadmium Absorption and Enrichment in Poplar Plants

**DOI:** 10.3390/ijms24010052

**Published:** 2022-12-20

**Authors:** Fang He, Qian Zhao, Yu-Jie Shi, Jun-Lin Li, Ting Wang, Tian-Tian Lin, Kuang-Ji Zhao, Liang-Hua Chen, Jia-Xuan Mi, Han-Bo Yang, Fan Zhang, Xue-Qin Wan

**Affiliations:** 1Sichuan Province Key Laboratory of Ecological Forestry Engineering on the Upper Reaches of the Yangtze River, College of Forestry, Sichuan Agricultural University, Chengdu 611130, China; 2College of Landscape Architecture, Sichuan Agricultural University, Chengdu 611130, China

**Keywords:** poplar, cadmium enrichment, UVB, phytoremediation, gene expression

## Abstract

The phenomenon of cross adaptation refers to the ability of plants to improve their resistance to other stress after experiencing one type of stress. However, there are limited reports on how ultraviolet radiation B (UVB) pretreatment affects the enrichment, transport, and tolerance of cadmium (Cd) in plants. Since an appropriate UVB pretreatment has been reported to change plant tolerance to stress, we hypothesized that this application could alter plant uptake and tolerance to heavy metals. In this study, a woody plant species, 84K poplar (*Populus alba* × *Populus glandulosa*), was pretreated with UVB and then subjected to Cd treatment. The RT-qPCR results indicated that the UVB-treated plants could affect the expression of Cd uptake, transport, and detoxification-related genes in plants, and that the UVB-Pretreatment induced the ability of Cd absorption in plants, which significantly enriched Cd accumulation in several plant organs, especially in the leaves and roots. The above results showed that the UVB-Pretreatment further increased the toxicity of Cd to plants in UVB-Cd group, which was shown as increased leaf malonaldehyde (MDA) and hydrogen peroxide (H_2_O_2_) content, as well as downregulated activities of antioxidant enzymes such as Superoxide Dismutase (SOD), Catalase (CAT), and Ascorbate peroxidase (APX). Therefore, poplar plants in the UVB-Cd group presented a decreased photosynthesis and leaf chlorosis. In summary, the UVB treatment improved the Cd accumulation ability of poplar plants, which could provide some guidance for the potential application of forest trees in the phytoremediation of heavy metals in the future.

## 1. Introduction

With the continuous increase in cadmium (Cd) input in the environment, the risk of Cd exposure faced by humans, animals, and plants is bound to become more and more serious [1,2]. Due to the rapid increase in industrial production and the rapid expansion of cities, the global annual consumption of Cd is 2 × 10^4^–2.4 × 10^5^ tons [3]. The main symptoms of plants suffering from Cd toxicity include a decrease in the photosynthetic rate, the inhibition of transpiration, a decrease in photosynthetic pigments, the chlorosis of plants, damage to the ultrastructure, the interruption of water and nutrient transport, the inhibition of growth and development, a decrease in biomass accumulation, protein hydrolysis, oxidative damage, and so on [4,5,6]. Plants under high Cd stress even appear to be wilting and have lesion necrosis. Cd^2+^ also suppresses electron transport chains in mitochondria and chloroplasts by replacing Fe^2+^ in plant proteins, resulting in reactive oxygen species (ROS) such as superoxide anions (O_2_^−^), hydroxyl radicals (·OH), and H_2_O_2_ [5,7].

In general, according to its properties, soil remediation technology can be divided into physical remediation, chemical remediation, and bioremediation [8]. Compared with the shortcomings of physical and chemical remediation methods such as tedious operation, high cost, and easy to cause secondary pollution, phytoremediation shows great advantages in soil remediation and has undoubtedly become the preferred remediation method for soil heavy metal pollution [9,10]. The defense modes of plants in response to Cd stress can be divided into four types: metal-resistant repulsive plants, metal-sensitive plants, metal-tolerant nonhyperaccumulator plants, and metal-tolerant hyperaccumulator plants [11]. When the content of Cd in the dry weight of plant leaves reaches 5–10 mg·kg^−1^, it will be toxic to most plants. However, in some Cd hyperaccumulators, even when the concentration of Cd reaches 100 mg·kg^−1^ in the tissues, the plants still show no negative symptoms [12]. At present, about 17 species of Cd hyperaccumulators have been identified, but most of them are herbaceous plants [13]. Poplar, as a fast-growing woody plant, has been proposed as an ideal species for phytoremediation due to its large biomass and strong ability to absorb Cd [14,15].

With the development of biotechnology and synthetic biology, the application of exogenous substances can significantly improve the tolerance of plants to metals and change their accumulation ability [16]. In addition to the protective mechanism of plants that can alleviate Cd toxicity, the cumulation and tolerance of plants to Cd can also be improved by adding trace elements, hormones, or acidic solutions to relieve the toxic response of plants to Cd stress [17]. For instance, when sodium hydrosulfide (NaHS) was applied, the content of soluble Cd in plants was significantly reduced [18]. At the same time, NaHS can also improve the antioxidant system and reduce the accumulation of H_2_O_2_ and lipid peroxidation by increasing the total activities of antioxidant enzymes (such as APX, CAT, and glutathione reductase (GR)) in plants [19]. In addition, other studies have shown that the addition of trace elements such as Zinc (Zn), Ferrum (Fe), and Manganese (Mn) can also interfere with the absorption of Cd by plants, thus reducing the concentration of Cd in plants [20]. On the contrary, spraying Silicon (Si) and Selenium (Se) on leaves can effectively increase the transport ability of Cd from stems to leaves [17].

Ultraviolet radiation B (UVB) refers to the light wave with a wavelength of 280–320 nm, which can damage plant DNA and induce the production of ROS, leading to the damage of the photosystem (PS II), and can even affect the tissue structure, morphological changes, and photosynthesis of plants [21]. Concurrently, UVB radiation can also induce a remarkable decrease in plant bioaccumulation and an increase in the content of phenolic pigments [22]. Moreover, when plants absorb UVB, a variety of phenolic compounds such as flavonoids and anthocyanins continue to accumulate in the cells to eliminate excessive ROS, thus alleviating the damage caused by UVB radiation in plants [23,24]. Furthermore, it has been pointed out that appropriate UVB radiation treatment can activate the defense mechanism of silver birch (*Betula pendula*) while enhancing plant resistance to drought stress [25]. Recently, the thinning of the ozone layer has led to a gradual increase in UVB radiation entering the Earth [26]. However, how UVB pretreatment affects the enrichment, transport, and tolerance of Cd in plants still remains unknown. Therefore, in this study, we explored how UVB pretreatment affects the physiological and molecular mechanisms of poplar plants under Cd stress. The results will provide some guidance for the potential application of forest trees in the phytoremediation of heavy metals in the future.

## 2. Results

### 2.1. Plant Growth Characteristics

It is possible that Cd exacerbated the morphological damage induced by the UVB treatment (Figure 1a–f). Firstly, the whole plant was dwarfed after treatment, and inhibition in the growth of the plant height was observed (Figure 1c,d). Secondly, the health degree of the leaves deteriorated, and the unhealthy characteristics of poplar seedlings such as the yellowing of leaves, decrease in the leaf area, and the number of leaves were observed. The yellow–brown spots were observed on the surface of the leaves under Cd stress (including the Cd group and UVB-Cd group) (Figure 1f), and the chlorosis of the leaves in the UVB-Cd group was the most serious. Through the comparison of the root phenotype (Figure 1e), it was found that the root system of the UVB-Cd group was significantly less developed than that of other treatment groups. Both treatments significantly inhibited plant height and a net increase in plant height (Figure 1g). Compared with the CK group, the net growth of plant height in the UVB, Cd, and UVB-Cd groups decreased by 14.32%, 16.94%, and 27.07%, respectively (Figure 1h). In short, through phenotypic analysis, it was found that the growth of plant height, leaves, and roots in the UVB-Cd group was inhibited and damaged. Moreover, there was no significant difference in biomass accumulation among plant tissues in the UVB group and CK group (Appendix A), which means that proper UVB treatment will not weaken plant growth. In addition, the dry weight of the leaves under the UVB-Cd treatment was significantly lower than that of the CK group (Appendix A). Compared with the control group, the root–shoot ratio increased significantly after the Cd treatment, and it was the largest in UVB-Cd group (Appendix A).

### 2.2. Analysis of Leaf Gas Exchange

To determine whether the UVB and Cd treatments affected plant growth through photosynthesis, the normal and treated groups were monitored for photosynthetic parameters (Figure 2). The parameters of the net photosynthetic rate (Pn) and stomatal conductance (Gs) of the poplar leaves were significantly different among different groups, and the changing trend was the same. Taking the CK group as a reference, the Pn of the UVB, Cd, and UVB-Cd groups (Figure 2a) decreased significantly by 30.95%, 57.80%, and 69.37%, respectively, and that of Gs (Figure 2d) decreased by 23.34%, 50.66%, and 79.24%, respectively. Except for the transpiration rate (Tr) of the UVB-Cd group, which was significantly lower than that of the CK group by 58.01%, the decrease in other treatment groups was not significant (Figure 2b). The intercellular CO_2_ concentration (Ci) parameters of the UVB and UVB-Cd groups (Figure 2c) were significantly different from those of the CK group, with a decrease of 16.08% and 25.13%, respectively. Therefore, after treatment, the photosynthesis of plants in the UVB-Cd group was severely restricted compared to the other treatments.

### 2.3. Oxidative Stress and Antioxidant Enzyme Activities

When plants encounter stress, the level of MDA, as an indicator of membrane peroxidation, will increase. We found that the MDA content of the Cd group and UVB-Cd group was 1.88 and 2.15 times higher than that of the CK group, respectively (Figure 3a). In addition, taking the CK group as a reference, Figure 3b shows that the content of H_2_O_2_ produced by the UVB, Cd, and UVB-Cd groups increased significantly, which was 1.26, 1.93, and 2.23 times higher than that of CK, respectively. The content of H_2_O_2_ produced by the UVB-Cd treatment was also significantly higher than that of the Cd treatment, with an increase of 15.70%.

At the same time, the accumulation of harmful substances can stimulate the plant antioxidant enzyme system, including SOD, CAT, and APX. The changes in the APX and CAT activities of the poplar leaves under different treatments were the same, which increased significantly compared to the CK group and had the highest value under the Cd treatment, followed by the UVB-Cd treatment, and showed the lowest under the UVB treatment (Figure 3c,d). Furthermore, the SOD activity of the poplar was 6.33, 5.14, and 4.75 times higher than that of the CK group after treatment with UVB, Cd, and UVB-Cd, respectively (Figure 3e). However, compared with the Cd group, the activities of APX, CAT, and SOD produced by the UVB-Cd group decreased by 11.48%, 18.95%, and 7.73%, respectively. Therefore, it can be concluded that the antioxidant capacity of the UVB-Cd group was lower than that of the Cd group.

### 2.4. Location and Distribution of Cd in Plants

Whether the change in plant resistance is determined by UVB or by Cd ion absorption needs further exploration. Through the histochemical staining of the plants, we could observe the complex precipitation of Cd- dithizone in the root, stem, and leaf tissues of the Cd group and UVB-Cd group. The red arrow highlights the distribution of Cd in various tissues and organs (Figure 4). It clearly showed that Cd was enriched in the top of the root (Figure 4(a3,a4)) and accumulated in the phloem of the stem (Figure 4(b3,b4)) and the leaf vein and mesophyll cells (Figure 4(c3,c4)). Through intuitive comparisons, we can find that the distribution density of Cd in each tissue of poplar in the UVB-Cd group was higher than that in the Cd group. This evidence suggests that UVB may alter the distribution and accumulation of Cd in plants.

### 2.5. Cd Content

To further determine whether UVB affects the distribution and accumulation of Cd in plants, we analyzed the content and total amount of Cd in plant tissues. The content of Cd in the tissues of poplar and soil from the CK and UVB groups was only at a trace amount (Appendix A), which was much lower than that from the Cd treatment (Cd and UVB-Cd group). In the follow-up study, we focused on comparing the enrichment and translocation of Cd by poplar between two Cd treatments (Cd group and UVB-Cd group). Cd accumulation in the poplar organs was consistent under the Cd group and UVB-Cd group. The accumulation of Cd in all tissues of the UVB-Cd group was the highest. In addition, in the UVB-Cd treatment group, the Cd enrichment in plant leaves and roots was significantly higher than that in the Cd group, which increased by 20.24% and 59.23%, respectively (Figure 5a).

Meanwhile, the shoot part, underground part, and total accumulation of plants in the UVB-Cd group were significantly increased by 21.43%, 59.23%, and 36.3% compared with those in the Cd group, respectively (Figure 5b). Furthermore, the concentration of Cd in the roots was much higher than that in other tissues (Figure 5c,d), and the content of Cd in poplar roots in the UVB-Cd group was even as high as 576.89 mg∙kg^−1^, which was 49.07% higher than that in the Cd group. Generally speaking, the total amount of Cd accumulated by poplars in the UVB-Cd group was higher than that in the Cd group. On the contrary, the Cd concentration of soil in the Cd group was much higher than that in the UVB-Cd group (Appendix A). Therefore, the total amount of Cd accumulated by poplars in the UVB-Cd group was higher than that in the Cd group. These results suggested that UVB pretreatment can enhance Cd absorption and enrichment in poplars.

### 2.6. Transcriptional Level of Genes Related to Cd Uptake, Transport, and Detoxification in Plants

To explore how UVB affects Cd uptake and enrichment in plants, we performed an RT-qPCR analysis on the genes related to Cd uptake, transport, and detoxification in the leaves and roots of the four groups (Figure 6 and Figure 7). Among them, the ARABIDOPSIS THALIANA ATP-BINDING CASSETTE C (*ABCC*), FLAVANONE 3-HYDROXYLASE (*F3H*), MYB DOMAIN PROTEIN (*MYB*), GLUTATHIONE SYNTHETASE (*GSH*), and PHOSPHORYLCERAMIDE SYNTHASE (*PCS*) protein families were responsible for the detoxification of heavy metals in plants. The NRAMP METAL ION TRANSPORTER (*NRAMP*), ZINC TRANSPORTER PRECURSOR (*ZIP*), METAL-TOLERANCE PROTEIN (*MTP*), CATION CALCIUM EXCHANGER (*CAX*), and YELLOW STRIPE-LIKE (*YSL*) protein families were responsible for the uptake and transport of Cd. The transcriptional level of these genes (*ABCC1*, *ABCC2*, and *GSH*) in the leaves of the UVB treatment were significantly upregulated compared with CK (Figure 6a,b,e). In addition, the *ABCC2* and *GSH* genes were significantly induced in the plant leaves of the Cd group (Figure 6b,e).

Moreover, the expression levels of three genes (*ABCC1*, *F3H,* and *MYB12*) were significantly upregulated in the leaves of the UVB-Cd group (Figure 6b–d). However, *PCS1* was significantly downregulated in the leaves of the three treatment groups, and the downregulation was most obviously in the UVB-Cd group (Figure 6f). There were six Cd transporter-related genes (*ZIP4*, *ZIP6*, *CAX2*, *CAX3*, *CAX6*, and *NRAMP1*) that were upregulated significantly in plant leaves after Cd exposure, whereas only two genes (*ZIP4* and *YSL1*) in the leaves were detected to be upregulated in the UVB-Cd group (Figure 6g–s).

In contrast, all the six Cd detoxification-related genes were significantly upregulated in the root of the UVB group, among which three genes (*F3H*, *MYB12*, *GSH*) were upregulated in the Cd group (Figure 7a–f). In addition to the *ABCC2* gene, the other five Cd detoxification-related genes were significantly induced in the roots of the UVB-Cd group. Furthermore, it was obviously found that most of the genes related to Cd transport were significantly upregulated in the roots of the Cd group and UVB-Cd group, but the expression levels of 10 genes (*ZIP2*, *ZIP4*, *CAX1-7,* and *NRAMP1*) in the roots of the UVB-Cd group were significantly higher than those in the roots of the Cd group (Figure 7g–s).

## 3. Discussion

### 3.1. UVB Pretreatment Affected Poplar Tolerance to Cd

Plants exposed to Cd are prone to morphological changes such as wilting and the yellowing of leaves, biomass reduction, and root growth inhibition [12]. The plant dry weight and seedling height of the poplar plants decreased significantly under the concentration of 100 umol·L^−1^ Cd^2+^ treatment. In accordance with our experimental results, the plants became shorter and their leaves became more yellow after the Cd treatment (Figure 1a–d). In addition, when the plants were exposed to UVB radiation for a certain amount of time and intensity, their height decreased [27]. After the UVB treatment in poplar trees, under the Cd treatment, the plants will dwarf and their leaves will turn yellow (Figure 1f–h). Leaf yellowing will weaken electron transfer during photosynthesis and result in photosynthesis inhibition, thus further affecting plant growth [28]. At the same time, Cd stress could lead to a decrease in stomatal conductance, which further weakens plant photosynthesis (Figure 2).

In addition, the effect of the UVB pretreatment can persist for at least 63 days. We found that there were significant differences in the indexes of photosynthesis and antioxidant capacity between the UVB pretreatment group and CK group after 63 days (Figure 2 and Figure 3).

However, there was no significant difference in plant root morphology and size after the UVB pretreatment (Figure 1c), which also resulted in the largest root–shoot ratio in the UVB-Cd group (Appendix A). The root–shoot ratio will increase gradually as the stress on the plant increases [29,30]. Moreover, plant height and biomass differences between the UVB-Cd and Cd treatment of the plants were small (Figure 1g,h and Appendix A). The reason for this is that the UVB pretreatment may be a signal, rather than a long-term stress, causing additional damage to the plant. In addition, UVB treatment can inhibit plant height, change the plant tissue structure, and thicken the spongy tissue and palisade tissue in the leaves to form small and thick leaves [31,32]. Therefore, the small difference in leaf biomass between the UVB-Cd group and Cd group may have been caused by the increase in leaf thickness despite the decrease in leaf size after the UVB pretreatment.

Furthermore, UVB stimulated plants to absorb more Cd (Figure 5b), further aggravating plant toxicity. Excessive Cd can produce a large number of ROS and MDA, leading to lipid peroxidation of the plant cell membrane, changing the structure of the cell membrane, and increasing the permeability of the plasma membrane [33]. The increase in the Cd enrichment in the plants after the UVB pretreatment resulted in the increase in ROS and MDA, which resulted in toxic effects on plants (Figure 3a,b). Excessive H_2_O_2_ and MDA activated the antioxidant system of the plants, including the enhancement of POD, APX, CAT, and SOD activities [34]. APX, CAT, and SOD play an important role in maintaining ROS homeostasis under heavy metal stress [35]. After the Cd stress, APX, CAT, and SOD activities in the poplar increased significantly. However, the enzyme activity of the UVB-Cd group was lower than that of the Cd group (Figure 3c–e), which might be the main reason for the decrease in Cd tolerance under the UVB pretreatment. Previous studies showed that with the intensification of heavy metal toxicity, the overall change trend of the activities of the antioxidant enzymes APX, CAT, and SOD increased first and then decreased [36]. When in a high-Cd-stress environment, the ability of plants to scavenge ROS is far less than the rate of ROS production, resulting in a large amount of ROS accumulation in plant tissues which will cause detrimental damage to plants.

### 3.2. UVB Alters Cd Uptake and Distribution in Poplar

For most plants, when the concentration of Cd under the leaf dry weight reaches 5–10 mg·kg^−1^, the plant will be poisoned by Cd stress [12]. However, the poplar leaves in this experiment, which had the highest concentration of Cd in the dry weight of the UVB-Cd group at 157.62 mg·kg^−1^ (Figure 5c), met one of the conditions of Cd super-enriched plants [37,38]. However, after the Cd treatment, the concentration and enrichment of Cd in poplar roots were significantly higher than those in the other tissues, and the UVB pretreatment significantly enhanced this result (Figure 5c,d). Roots, after all, are the first organ to come into contact with heavy metal ions in the environment and enable the transport of metals to other organs through the complex vascular tissues of the plant [39]. This was also confirmed by the presence of a large number of Cd ions in the leaf vein (Figure 4c). In addition, as a perennial woody plant, poplar’s root system has a strong degree of lignification and can store a lot of materials. Moreover, unlike annual plants (such as rice and *Arabidopsis thaliana*), poplar roots do not die directly when they mature. Although the roots of plants are not easy to harvest, the roots of poplars can absorb and store heavy metal ions for a long time. Many studies have found that poplar can play an important role as an excellent heavy metal remediation plant [40,41]. However, poplar is a perennial deciduous plant [40], so if the deciduous leaves are not artificially treated, the Cd accumulated in its leaves will return to nature. Therefore, if more Cd can be stored in poplar stems and roots and other organs, using poplar to absorb Cd will be a more economical and convenient strategy. Without affecting the biomass accumulation of the leaves (Appendix A), a slight increase in the Cd concentration of the leaves in the UVB-Cd group may have been the main factor for the large amount of Cd accumulation in the leaves of the UVB-Cd group compared to the Cd treatment group (Figure 5a,c). On the contrary, leaves are the main organs of plant photosynthesis, in which a large amount of Cd accumulation will also affect plant growth. Therefore, there is a balanced relationship between Cd accumulation and growth in plants [42]. Moreover, proper UVB is an emergency signal, rather than permanent damage to the plants caused by prolonged stress [43]. In the follow-up work, it is necessary to determine a threshold range of the UVB pretreatment to significantly increase the total Cd accumulation in plants, which has practical value for the phytoremediation of Cd pollution in the environment.

When wild sunflowers, mustard, mung beans, and clovers respond to Cd stress, similar results were found, which showed that the plant roots intercepted and enriched most of the Cd, alleviating the damage of the Cd to the upper tissue of the ground [10,44,45,46]. This indicated that poplar plants adopted a passive tolerance strategy to cope with a high concentration of Cd stress since they cannot block Cd ions from the outside of the plant or expel them from the body. In addition, UVB pretreatment enhanced this strategy and increased the overall Cd enrichment capacity of the plants by 36.3% (Figure 5c). Although proper UVB pretreatment will not affect plant biomass accumulation and make poplars absorb more Cd ions, it still takes a long time to be used in phytoremediation.

### 3.3. UVB Pretreatment Influence of Cd Uptake, Transport, and Detoxification in Plants

The aboveground part of the plants is the first organ to contact the UVB signal, and the root is the first organ to contact heavy metals [47]. How plants perceive UVB signals and then improve their Cd absorption capacity and change their plant tolerance needs to be further revealed from the molecular level. Moreover, the signal (UVB pretreatment group) could also activate the expression of the heavy metal detoxification genes (*ABCC1*, *ABCC2*, and *GSH*) and the antioxidant enzyme system (Figure 3c–e and Figure 6a,b,e) so as to alleviate the additional toxicity caused by the excessive absorption of Cd.

It has been documented that *ABCC1/2*, as a major member of the ABC transporter, plays an important role in plant detoxification and tolerance to heavy metals [48,49,50]. In Arabidopsis, these two proteins (*ABCC1/2*) are mainly present on the vacuole membrane and are involved in the isolation of heavy metals in vacuoles, enhancing plant tolerance [51]. Consistent with our results, the *ABCC1 and ABCC2* gene was significantly induced to participate in alleviating the toxic effects of Cd in the leaves of UVB-Cd and Cd groups, respectively (Figure 6a,b). In addition, both *MYB12* and *F3H* promote the accumulation of anthocyanins, which can improve plant tolerance to abiotic stress [52,53,54]. We found that the two genes in the leaves were significantly induced in the UVB-Cd group while the two genes in the roots were both induced in the UVB-Cd and Cd group, and the upregulation multiple was significantly higher than that in the leaves (Figure 7c,d). Furthermore, the *PCS* gene confers tolerance to Cd [55] and catalyzes phytochelatin synthesis from glutathione (*GSH*) in the presence of Cd^2+^, Zn^2+^, Cu^2+^, and Fe^3+^, but not Co^2+^ or Ni^2+^ [56]. Both *GSH* and *PCS1* genes in the poplar roots were induced in the UVB-Cd and Cd group roots (Figure 7e,f), which may be related to the accumulation of Cd ions in the roots.

Under different treatments, the expression differences of these detoxification-related genes in roots and leaves led to the difference in Cd tolerance. The *NRAMP*, *ZIP*, *MTP*, *CAX*, and *YSL* protein families are responsible for the uptake and transport of Cd in plants [57,58,59]. On the whole, the expression levels of most genes were induced to a higher degree in the roots than in the leaves after treatment, which also resulted in the retention of most Cd ions in the roots. It has been documented that *Zip2/4* in *Arabidopsis* is significantly induced by Cd and is involved in Cd transport and absorption [60]. Moreover, the *CAX* family in poplar and *Arabidopsis* plays an important role in Cd enrichment and tolerance [61,62,63]. In addition, *NRAMP1* is located on the plasma membrane and can significantly promote the absorption and accumulation of heavy metals in rice [64,65]. After the UVB-Cd treatment, the expression levels of the ten genes (*ZIP2/4*, *CAX1/2/3/4/5/6/7*, and *NRAMP1*) in the roots were significantly induced and were much higher than those in the Cd group, which may be the main reason for the higher uptake of Cd in the UVB-Cd group (Figure 7). Similarly, nine genes (*ZIP2/4/6*, *CAX2/3/5/6/7*, and *NRAMP1*) were significantly upregulated in the root after the UVB treatment, which further suggests that UVB pretreatment can activate the expression of these metal transporters and promote the uptake of Cd ions in plants. UVB pretreatment and Cd treatment coaffect the expression of metal-transport-related genes to control Cd uptake and transport in plants.

## 4. Materials and Methods

### 4.1. Plant Material

The tissue culture seedlings of 84K poplar (*Populus alba* × *Populus glandulosa*) were collected from the Forest cultivation Laboratory of Sichuan Agricultural University in Chengdu, Sichuan, and were cultured in a tissue culture room with an ambient temperature of 22 ± 2 °C, light 16 h, and dark 8 h (light time of 6:00–22:00) [66].

In the early stage of the experiment, a 1/2 MS medium (caisson, Smithfield, UT, USA) was selected for the mass propagation of the 84K poplar tissue culture seedlings in the tissue culture room. After growing for about 30 days, the seedlings with consistent growth and healthy growth were selected to transplant into the same size pots (7 cm × 5 cm × 7.7 cm) which were filled with soil (peat soil: vermiculite: perlite = 2:2:1) to continue cultivation in the culture room (temperature 22 ± 2 °C, 16 h light, and 8 h darkness).

### 4.2. Poplar UVB Pretreatment and Cd Treatment

After 15 days of culture, the 84K poplar seedlings with the same growth were selected and subjected to the following four treatment groups: control group, UVB group, Cd group, and UVB-Cd group. Each group contained 9 plants for replication. To analyze the stimulation of UVB stress on the defense mechanism of the 84K poplar and thus affect the Cd absorption and tolerance ability of plants, we referred to previous research methods [40,67] with modifications. Due to the difference in plant species, we modified the number and frequency of UVB pretreatments and the concentration of Cd treatments. The specific treatment methods were as follows: On the 0–8th day, the UVB pretreatment was performed, and the ultraviolet lamp (UVB) was fixed on the top of the poplar seedlings. The time of each UVB treatment was at the same time (9:00 a.m.–9:30 a.m.) of the day, and the processing frequency was 0.5 h/times/3 days, for a total of 3 times. From the 9th to the 63rd day, the Cd treatment was carried out. The stock solution of 2 mmol/mL concentration was first prepared with CdCl_2_·2.5H_2_O (CHRON CHEMICALS, Qionglai, China). After taking 100 μL of stock solution to 1 L each time, each poplar was treated with 100 mL of the Cd solution at the same time every 3 days, that is, the treatment frequency was 20 μmol/time/3 days for a total of 19 times. At the same time, all the seedlings were watered daily to prevent drought.

The seedlings in the control group without any treatment during the experiment were recorded as the CK group, the seedlings only involved in the UVB pretreatment stage were recorded as the UVB group, the seedlings that only participated in the Cd treatment stage were recorded as the Cd group, and the seedlings in the UVB pretreatment and Cd treatment stages were recorded as the UVB-Cd group. Among them were the number, frequency, and grouping of the UVB and Cd processing, as shown in Appendix A.

### 4.3. Measurement of Plant Parameters

#### 4.3.1. Determination of Growth Index

During the experiment, the plant height of each poplar was measured with a steel tape measure (precision 0.1 cm) every 9 days, and the net growth of the plant height was calculated by subtracting the plant height at the end of the experiment from the initial plant height. There were 9 biological repeats in each treatment, and three more technical repeats were carried out in each biological repeat. The total number of repeats per treatment = biological repeats × technical repeats.

After the end of the treatment, the leaf, stem, and root tissues of each poplar were harvested separately before being packed separately in paper bags and put into an oven for 30 min at 105 °C followed by 7 days at 70 °C to a constant weight [68,69]. The dry weight of the plant tissues under different treatments was weighed and recorded for the calculation of the root to shoot ratio (root dry weight/aboveground dry weight).

#### 4.3.2. Determination of Photosynthetic Parameters

The 5th–7th healthy mature leaves of poplar from top to bottom were selected, and a Li-6800 portable photosynthesis-fluorescence analyzer (Li-Cor, Lincoln, NE, USA) was used [70]. According to the previous research method [39], the photosynthetic active radiation was 800 μmol∙m^−2^∙s^−1^, the concentration of CO_2_ was 400 μmol·mol^−1^, the temperature was 22 ± 2 °C, and the time 9:00–17:00 was when we measured the photosynthetic index. The net photosynthetic rate (Pn), transpiration rate (Tr), stomatal conductance (Gs), and concentrations of intercellular CO_2_ (Ci) of each plant were measured. Nine plants were selected as replicates for each treatment, and three leaves were selected for each plants.

#### 4.3.3. Determination of Physiological Indicators

Without the influence of external factors, the physiological indexes of the mature leaves (functional leaves) of poplar are relatively stable [71]. The 5th–7th healthy mature leaves of each plant from the four treatment groups were sampled and quickly placed in the ice box. Then, the physiological and biochemical indexes were measured in the laboratory immediately. A total of 0.2 g of each leaf sample was weighted and ground with 1.5 mL of the phosphate buffer. The homogenized liquid was transferred to a 5 mL tube for the determination of malonaldehyde (MDA), hydrogen peroxide (H_2_O_2_), Superoxide Dismutase (SOD), Catalase (CAT), and Ascorbate peroxidase (APX). The content of MDA was determined using the Thiobarbituric acid method, and the absorption values at 600 nm, 532 nm, and 450 nm wavelengths were recorded [72]. The contents of H_2_O_2_, SOD, CAT, and APX were determined with the Hydrogen Peroxide assay kit, Total Superoxide Dismutase (T-SOD) assay kit (Hydroxylamine method), Catalase (CAT) assay kit (Visible light), and Ascorbate peroxidase (APX) test kit (Nanjing Jian cheng Biological Engineering Institute, Nanjing, China), respectively. There were 9 biological replicates and 3 technical replicates for each treatment.

#### 4.3.4. Histochemical Staining of Poplar Tissues

At the end of the experiment, the tissues of each plant were washed with deionized water, and the fresh tissues were taken from the same parts of the leaves, stems, and roots of the poplars in each group. Transverse sections of the 8th internode of the poplar were obtained using a semiautomatic frozen slicer. Then, all tissues and organs were stained with a Cd staining solution (mixed with 60 mL of acetone, 2 mL of deionized water, and 100 mL of glacial acetic acid, and then 30 mg of diphenylthiocarbazone were added). After dyeing for 1 h, the tissues were rinsed with deionized water 2–3 times until the residual dye on the tissue surface was clean. Since diphenylthiocarbazone reacts with Cd^2+^ to form a red–black complex, we used an optical microscope (BX51, Olympus, Tokyo, Japan) to observe and photograph the adsorbed Cd^2+^ in the plant tissue [40]. At least 100 photographs were taken of each tissue site for each treatment.

#### 4.3.5. Determination of Cd in Plant and Soil Samples

The dried plants were crushed and screened (0.15 mm) according to the leaf, stem, and root tissue. The soil samples were ground and screened in each plant. A total of 0.2 g of screened plant and soil samples were digested using the wet method [73]. Finally, inductively coupled plasma mass spectrometry (ICP-MS) was used to determine the content of Cd in plants and soil. Before the test, we soaked all the test utensils, including a funnel, anticooking tube, volumetric bottle, etc., with 5% HNO_3_ overnight, rinsed them with deionized water many times before use, and left them to dry naturally. The sample was sent directly to the bottom of the dry digestion tube through a long spoon to prevent the sample from remaining on the wall of the digestion tube. In total, 12 mL of the mixed acid (concentrated nitric acid: perchloric acid = 5:1) was added vertically to the digestion tube, and we also paid attention to avoiding residual drugs on the wall of the tube. Then, we put the leak on the digestion tube and placed it in the ventilation cupboard for the night. Then, the digestion tube was placed on the graphite digestion meter, and the temperature stage was set to 80 °C (20 min), 120 °C (20 min), 160 °C (20 min), and 180 °C (60 min) [73]. After digestion, the clear digestion solution cooled to room temperature was transferred to a capacity bottle of 50 mL, the digestion tube was washed with deionized water many times, and then the liquid was fixed to a volume of 50 mL. We filtered the tested liquid into a clean polyethylene bottle with filter paper to determine the content of Cd on the ICP-MS. First, according to the operational requirements of the test instrument, the Cd standard was configured with concentration gradients of 0, 50, 100, 150, 200, 250, 300, 350, and 400 μg·L^−1^. The Cd content was determined using ICP-MS (NexION1000G, PerkinElmer, Suzhou, China) sequentially. After obtaining the standard curve, the Cd content in the sample was determined, and the total Cd content in each basin was calculated according to the dry weight of each tissue. There were 9 biological repeats and 3 technical repeats in each treatment.

#### 4.3.6. RNA Extraction and RT-qPCR Analysis

At the end of the treatment, the leaves and roots of poplars in the same part of each group were sampled and wrapped in tin foil and quickly put into liquid nitrogen. Then, the tissue parts were ground to powder with liquid nitrogen in a mortar without an RNA enzyme. The total RNA of the leaves and roots was extracted using the plant total RNA extraction kit (TSP412, Tsingke, Beijing), and the cDNA was synthesized using the reverse transcription kit (TSK314S, Tsingke, Beijing). Referring to previous studies, the related genes responding to Cd stress were selected, and *PtrActin* and *PtrUBQ* genes were used as internal reference genes for RT-qPCR analysis [74]. Five biological replicates were performed at each time, and four technical replicates were performed for each sample. The PCR was operated on Bio-Rad CFX96 equipment (Bio-Rad, Hercules, CA, USA) and the 2-delta Ct algorithm was used to analyze the results [75]. All the primers were designed using NCBI and DAMAN and were synthesized by Optimus Biotechnology (Co., Ltd., Xiamen, China) (Appendix A).

### 4.4. Statistical Analyses

SPSS 27.0 (SPSS, Chicago, IL, USA) was used to standardize the data and test the significant difference, and a one-way analysis of variance (one-way ANOVA) was selected and compared using the Duncan method. In addition, the Student’s *t*-test was used to analyze the significance. Finally, GraphPad Prism7 (GraphPad Software, San Diego, CA, USA) was used to draw the statistical chart.

## 5. Conclusions

The data acquired from this study were used to construct a working model to explore the UVB-pretreatment-enhanced Cd absorption and enrichment in poplar (Figure 8). After the UVB pretreatment, the expression of Cd-uptake-, transport-, and detoxification-related genes in plants was stimulated and promoted the absorption of more Cd in poplar trees. Concurrently, more Cd ions were enriched in poplar leaves. This also led to a burst of MDA and H_2_O_2_ in the UVB-Cd group leaves, resulting in reduced leaf photosynthesis and leaf chlorosis. In general, proper UVB pretreatment can enhance Cd accumulation in poplar plants. This result will improve the Cd accumulation efficiency of poplars and provide some guidance for the application of forest trees in the phytoremediation of heavy metals in the future.

## Figures and Tables

**Figure 1 ijms-24-00052-f001:**
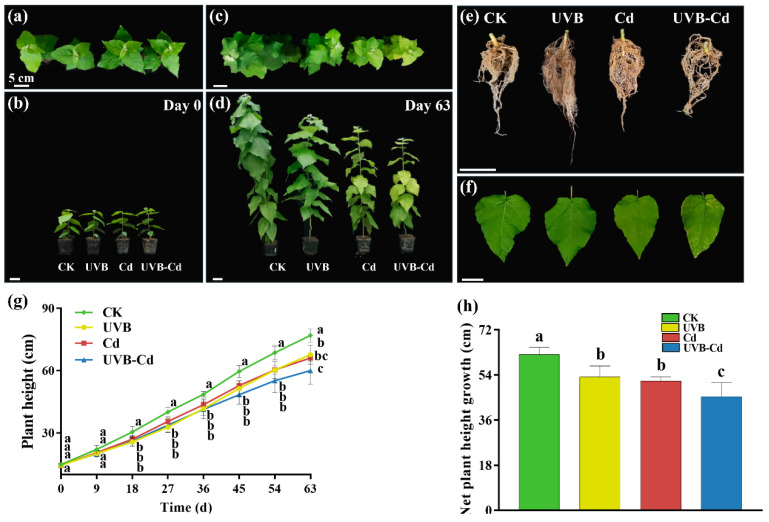
Phenotypic differences of the poplar before and after treatment. (**a**–**d**) T of the whole plant. (**e**) Root and (**f**) leaf phenotypes of poplar after treatment. (**g**) Plant height. (**h**) Net plant height growth (*n* = 9 plants for each treatment). Different letters represent significant differences among four treatment groups (*p* < 0.05).

**Figure 2 ijms-24-00052-f002:**
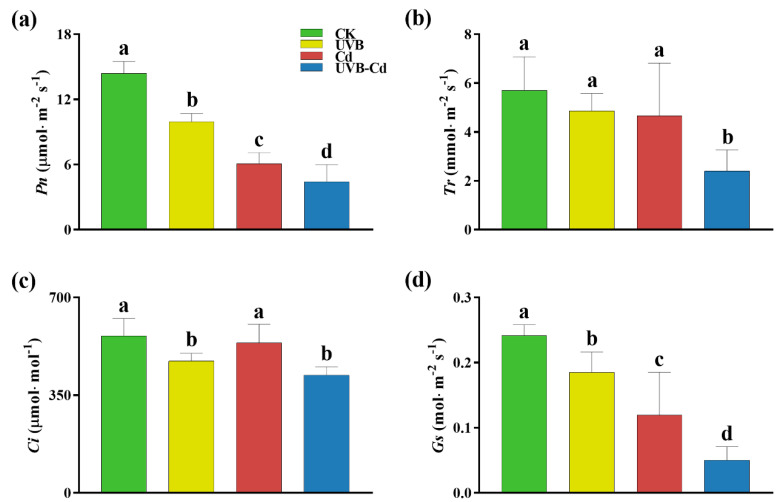
The photosynthetic gas exchange parameters of poplar under different treatments. (**a**) Net photosynthetic rate (Pn). (**b**) Transpiration rate (Tr). (**c**) Intercellular CO_2_ concentration (Ci). (**d**) Stomatal conductance (Gs). Different letters represent significant differences among four treatment groups (*p* < 0.05, *n* = 9 plants for each treatment).

**Figure 3 ijms-24-00052-f003:**
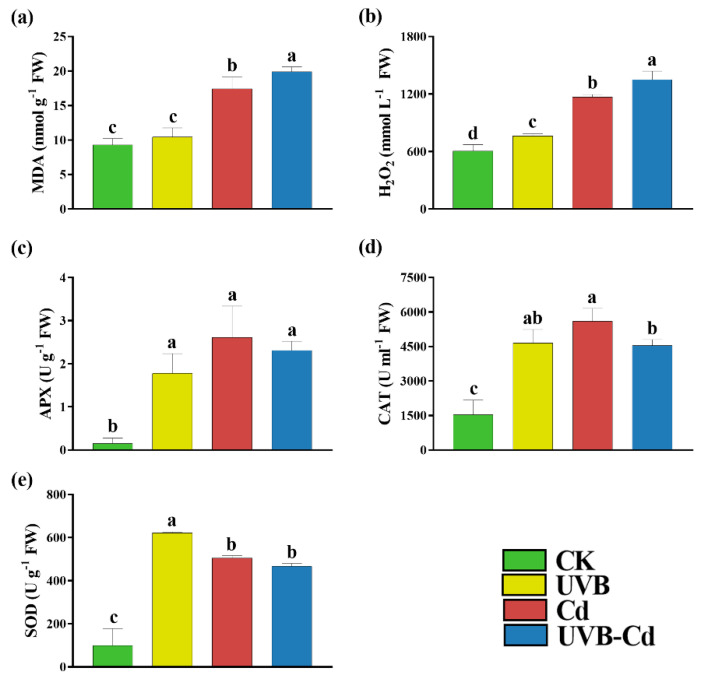
Indicators related to the antioxidant capacity of 84K poplar leaves in different treatment groups. The content of MDA (**a**) and H_2_O_2_ (**b**), APX (**c**), CAT (**d**), and SOD (**e**) of poplar leaves in different treatment groups (*n* = 9 plants for each treatment). Different letters represent significant differences among four treatment groups (*p* < 0.05).

**Figure 4 ijms-24-00052-f004:**
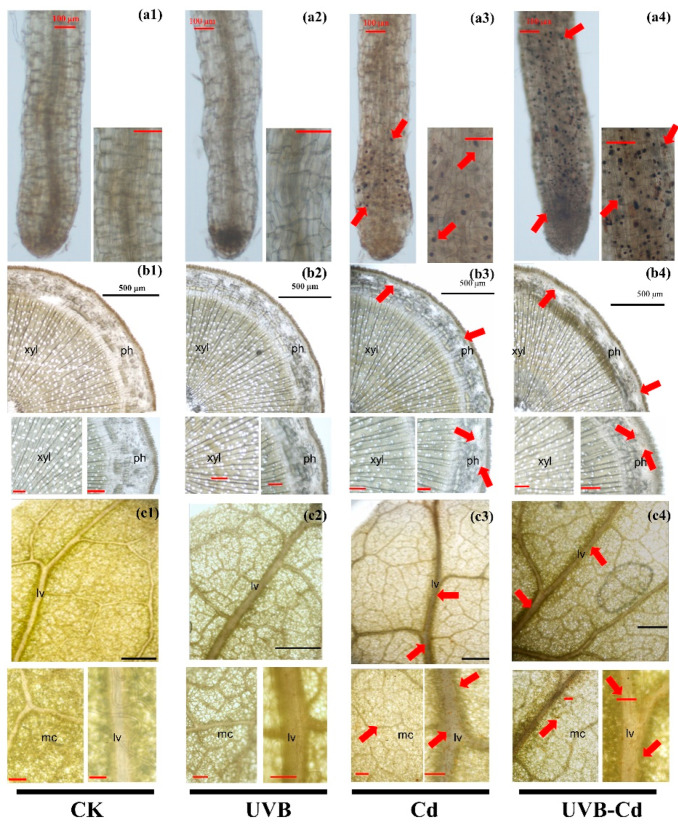
Cd localization and accumulation in roots (**a1**–**a4**), stems (**b1**–**b4**) and leaves (**c1**–**c4**) of poplar under different treatments. The red arrow indicates the precipitation of Cd-dithizone. Red bar scale = 100 μm, black bar scale = 500 μm. ph, phloem; xyl, xylem; lv, leaf vein; mc, mesophyll cells. The experiment was repeated at least three times, each time with similar results. A group of photographs was selected for display.

**Figure 5 ijms-24-00052-f005:**
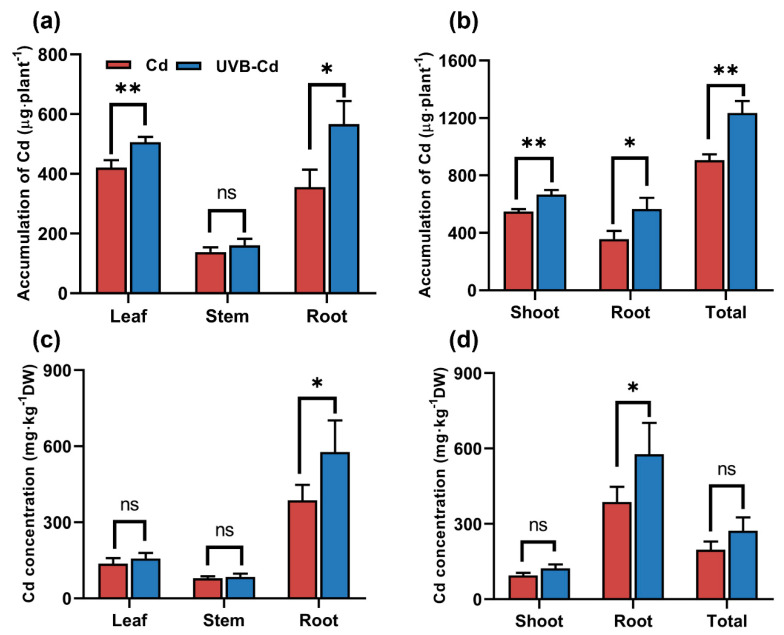
The total accumulation of Cd in different tissues (**a**) and the whole plant (**b**), and the Cd content in different tissues (**c**) and the whole plant (**d**) under different treatments (*n* = 9 plants for each treatment). Asterisks represent significant differences between two treatment groups: * represents *p* < 0.05; ** represents *p* < 0.01.

**Figure 6 ijms-24-00052-f006:**
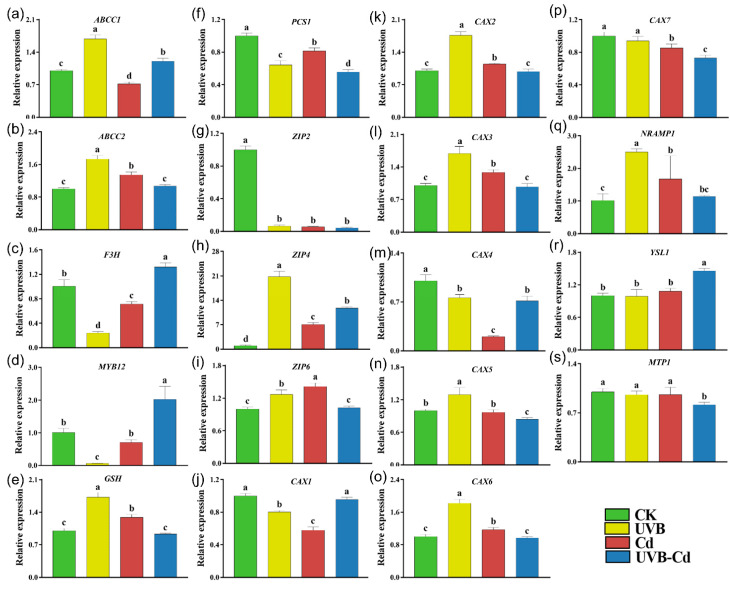
Relative expression of genes related to Cd uptake, transport, and detoxification in poplar leaf under different treatments. The expression levels of *ABCC1* (**a**), *ABCC2* (**b**), *F3H* (**c**), *MYB12* (**d**), *GSH* (**e**), *PCS1* (**f**), *ZIP2* (**g**), *ZIP4* (**h**), *ZIP6* (**i**), *CAX1* (**j**), *CAX2* (**k**), *CAX3* (**l**), *CAX4* (**m**), *CAX5* (**n**), *CAX6* (**o**), *CAX7* (**p**), *NRAMP1* (**q**), *YSL1* (**r**), and *MTP1* (**s**) in poplar leaf (*n* = 5 plants for each treatment). Different letters represent significant differences among four treatment groups (*p* < 0.05).

**Figure 7 ijms-24-00052-f007:**
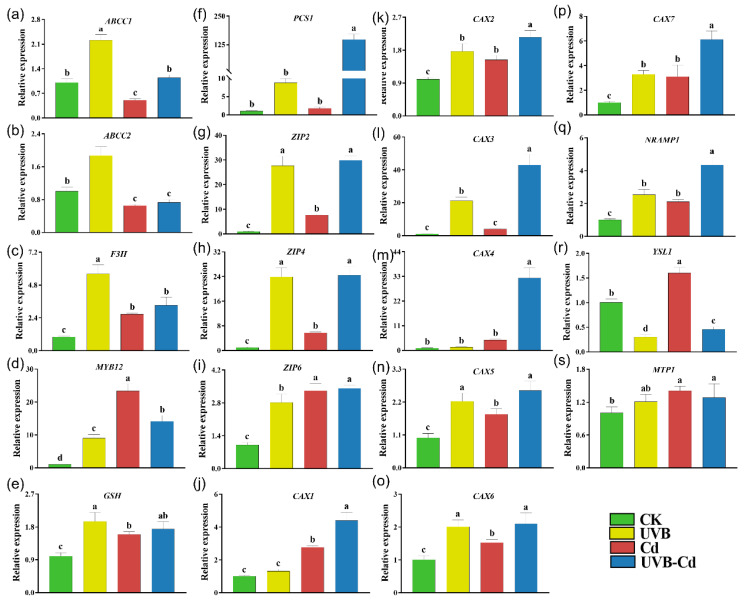
Relative expression of genes related to Cd uptake, transport, and detoxification in poplar root under different treatments. The expression levels of *ABCC1* (**a**), *ABCC2* (**b**), *F3H* (**c**), *MYB12* (**d**), *GSH* (**e**), *PCS1* (**f**), *ZIP2* (**g**), *ZIP4* (**h**), *ZIP6* (**i**), *CAX1* (**j**), *CAX2* (**k**), *CAX3* (**l**), *CAX4* (**m**), *CAX5* (**n**), *CAX6* (**o**), *CAX7* (**p**), *NRAMP1* (**q**), *YSL1* (**r**), and *MTP1* (**s**) in poplar root (*n* = 5 plants for each treatment). Different letters represent significant differences among four treatment groups (*p* < 0.05).

**Figure 8 ijms-24-00052-f008:**
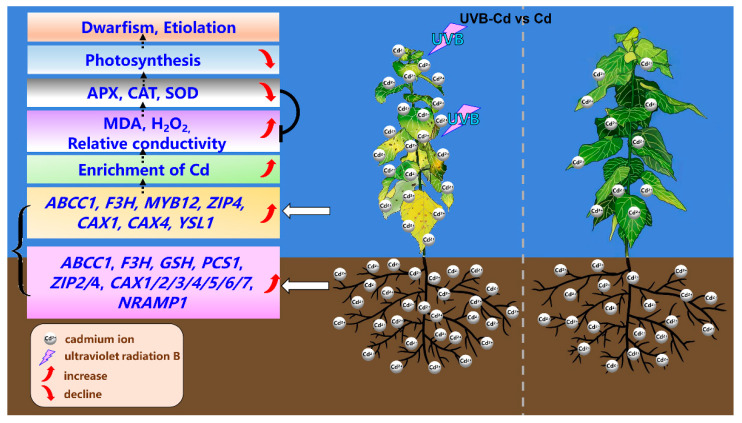
Physiological and molecular mechanisms of UVB Pretreatment enhanced Cd absorption and enrichment in poplar.

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
