# Peer review of "UVB-Pretreatment-Enhanced Cadmium Absorption and Enrichment in Poplar Plants"

_ijms, 2022, doi:10.3390/ijms24010052_

Round 1
Reviewer 1 Report (New Reviewer)
The phenomenon of cross-adaptation refers to the ability of plants to improve their resistance to other stress after experiencing one type of stress. However, there are limited reports on
how ultraviolet radiation B (UVB) pretreatment affects the enrichment, transport, and tolerance of cadmium (Cd) in plants. The authors found that the UVB pretreatment can enhance Cd accumulation in poplar tissues. This method
will improve the Cd accumulation ability of poplar plants, as well as provide some guidance for the application of forest trees in the phytoremediation of heavy metals in the future.
In general, the manuscript is well wirtten, whereas some places remain to be improved.
1. L33, 20000-240000 tons, please correct the number into the scientific notation form.
2. In the section 2.3.3, the abbreviations should be placed after their first spelling of full names.
3. Some spaces are missing, please check them throughout the manuscript. For example, L129.
4. According to the growth performance of polar plants under Cd and UVB+Cd,UVB seems to inhibit the growth of polar plants, including both leaves and roots.
Therefore, the leaf weights should be decreased after the plants exposure to UVB+Cd than to Cd. In Figure 5, the leaf Cd concentrations are similar between the two treatments.
Therefore, I am wondering that why the leaf Cd content/accumualtion is increased under UVB+Cd than under Cd. Cd accumulation/content= Dry weight* Cd concentration.
5. In the main text and figure legends, the gene names should be presented in italics.
6. UVB+Cd had a more pronounced inhibitory effect on the growth of polar plants that Cd, although UVB+Cd increased the polar Cd accumulation. Therefore, what are the values of this study in the practical production.
7. The language should be improved throughout the manuscript.
Author Response
Dear Editor:
We would like to thank you and the Reviewers for your valuable comments and suggestions that helped us further our understanding of some aspects on this field and to improve our manuscript. After carefully considering the Reviewers’ comments, we revised our manuscript by point-by-point, and hereby list our responses to these comments.
Due to the journal word limit, some methodological steps were simplified in our first manuscript. To address your concerns, we have altered the manuscript to include more detailed methods and materials, and more appropriate references, more adequate data interpretation. The manuscript was sent to a native English speaker for correct the text.
Please do not hesitate to contact me in case the manuscript has any further issues.
Thank you very much for your help.
We are looking forward to hearing from you soon.
Yours sincerely,
Fang He, Ph.D.
Associate professor of forest tree genetics and breeding
College of Forestry,
Sichuan Agricultural University , Chengdou 611130 , China
Tel: +86-17683777884
E-mail: 327187269@qq.com / 14686@sicau.edu.cn
-------------------------------------------------------------------------------------------------------
Responses to reviewer 1:
【Comment 1】1. L33, 20000-240000 tons, please correct the number into the scientific notation form.
[Our response1]: Thank you for your advice. We have changed this number to scientific notation form.
As nicely advised, the 'Introduction' section update in this paper.
The original is “Due to the rapid increase … of Cd is 20000-240000 tons [3].”, and we have revised it to “Due to the rapid increase in industrial production and the rapid expansion of cities, the global annual consumption of Cd is 2×104 – 2.4×105 tons [1].” (in line33-34).
【Comment 2】In the section 2.3.3, the abbreviations should be placed after their first spelling of full names.
[Our response 2]: Thank you for your advice and question. We have added the full names of all abbreviations in the full text.
【Comment 3】Some spaces are missing, please check them throughout the manuscript. For example, L129.
[Our response 3]: Thank you for your advice. We have carefully reviewed the full text and corrected these errors.
【Comment 4】According to the growth performance of polar plants under Cd and UVB+Cd,UVB seems to inhibit the growth of polar plants, including both leaves and roots. Therefore, the leaf weights should be decreased after the plants exposure to UVB+Cd than to Cd. In Figure 5, the leaf Cd concentrations are similar between the two treatments. Therefore, I am wondering that why the leaf Cd content/accumualtion is increased under UVB+Cd than under Cd. Cd accumulation/content= Dry weight* Cd concentration.
[Our response 4]:
Thank you for your question. According to the dry weight measurement of plant tissues, there was no significant difference in the dry weight of leaves and roots in the Cd group and the UVB-Cd group, respectively (Fig S2a).
In the discussion, we gave an in-depth explanation of the comment. As the reviewers saw, UVB seemed to inhibit plant growth. However, biomass differences between UVB-Cd and Cd treatment of plants were small (Fig 1g-h and Fig S2a). The reason for this is that UVB pretreatment may be a signal, rather than a long-term stress. In addition, UVB treatment can inhibit plant height, change plant tissue structure, thicken spongy tissue and palisade tissue in leaves to form small and thick leaves[2, 3]. Therefore, the small difference in leaf biomass between UVB-Cd group and Cd group may be caused by the increase in leaf thickness despite the decrease in leaf size after UVB pretreatment. Without affecting the biomass accumulation of leaves (Fig.S2a), a slight increase in the cadmium concentration of leaves in the UVB-Cd group may be the main factor for the large amount of cadmium accumulation in the leaves of the UVB-Cd group compared to the Cd treatment group (Fig.5a and c).
As nicely advised, the discussion update in this paper.
“In addition, UVB treatment can inhibit plant height, change plant tissue structure, thicken spongy tissue and palisade tissue in leaves to form small and thick leaves[40, 41].Therefore, the small difference in leaf biomass between UVB-Cd group and Cd group may be caused by the increase in leaf thickness despite the decrease in leaf size after UVB pretreatment.” (in line406-410).
“Without affecting the biomass accumulation of leaves (Fig.S2a), a slight increase in the cadmium concentration of leaves in the UVB-Cd group may be the main factor for the large amount of cadmium accumulation in the leaves of the UVB-Cd group compared to the Cd treatment group (Fig.5a and c)。” (in line 448-452).
【Comment 5】In the main text and figure legends, the gene names should be presented in italics.
[Our response 5]: Thank you for your question. We have changed the names of all the genes in the text to italics.
【Comment 6】UVB+Cd had a more pronounced inhibitory effect on the growth of polar plants that Cd, although UVB+Cd increased the polar Cd accumulation. Therefore, what are the values of this study in the practical production.
[Our response 6]: Thank you for your question. We have discussed this issue in the paper. For plants, as long as UVB pretreatment can increase the enrichment of cadmium in plants, we can use this way to repair cadmium pollution in the environment. Many studies have found that poplar can play an important role as an excellent heavy metal remediation plant[4, 5]. However, poplar is a perennial deciduous plant[4], if not artificially treated deciduous, the Cd accumulated in its leaves will return to nature. Therefore, if more cadmium can be stored in poplar stems and roots and other organs, it will be more economical and convenient strategy. Without affecting the biomass accumulation of leaves (Fig.S2a), a slight increase in the cadmium concentration of leaves in the UVB-Cd group may be the main factor for the large amount of cadmium accumulation in the leaves of the UVB-Cd group compared to the Cd treatment group (Fig.5a and c). On the contrary, leaves are the main organs of plant photosynthesis, in which a large amount of cadmium accumulation will also affect plant growth. Therefore, there is a balanced relationship between cadmium accumulation and growth in plants[6]. Moreover, proper UVB is an emergency signal, rather than permanent damage to plants caused by prolonged stress [7]. In the follow-up work, it is necessary to determine a threshold range of UVB pretreatment to significantly increase the total cadmium accumulation in plants, which has practical value for phytoremediation of cadmium pollution in the environment. Although proper UVB pretreatment will not affect plant biomass accumulation and make poplars absorb more cadmium ions, it still takes a long time to be used in phytoremediation.
As nicely advised, the discussion update in this paper.
However, poplar is a perennial deciduous plant[28], if not artificially treated deciduous leaves, the Cd accumulated in its leaves will return to nature. Therefore, if more cadmium can be stored in poplar stems and roots and other organs, it will be more economical and convenient strategy. Without affecting the biomass accumulation of leaves (Fig.S2a), a slight increase in the cadmium concentration of leaves in the UVB-Cd group may be the main factor for the large amount of cadmium accumulation in the leaves of the UVB-Cd group compared to the Cd treatment group (Fig.5a and c). On the contrary, leaves are the main organs of plant photosynthesis, in which a large amount of cadmium accumulation will also affect plant growth. Therefore, there is a balanced relationship between cadmium accumulation and growth in plants[49]. Moreover, proper UVB is an emergency signal, rather than permanent damage to plants caused by prolonged stress [50].In the follow-up work, it is necessary to deter-mine a threshold range of UVB pretreatment to significantly increase the total cadmium accumulation in plants, which has practical value for phytoremediation of cadmium pollution in the environment.” (in line 444-459).
【Comment 7】7. The language should be improved throughout the manuscript.
[Our response 7]: Thank you for your advice. We have invited native English speakers to help us polish the papar.
Thank you for your comments. All questions are answered in the text.
Responses to reviewer 2:
【Comment 1】Line 104: The modifications that authors employed different from the previous research methods should be stated to enable readers and other researchers understand such modifications in the methods.
[Our response1]: Thank you for your suggestion. In order to enable readers and other researchers to understand the modification of this method, we have supplemented the content related to the modification of the treatment method in the materials and methods.
The original is “To analyze the stimulation of UVB stress … with modification. ”, and we have revised it to “To analyze the stimulation of UVB stress on the defense mechanism of 84K poplar, and thus affect the plant's ability to absorb and tolerate Cd, we referred to previous research methods [4, 8] with modification. Due to the difference of plant species, we modified the number and frequency of UVB pretreatment and the concentration of Cd treatment. The specific treatment methods are as follows.” (in line 106-108)
【Comment 2】Line 126: What do authors mean by technical repeats in each treatment?
[Our response 2]: Thank you for your question. In order to eliminate the instrumental effects on data measurement, we have 9 biological repeats in each treatment, and three more technical repeats have been carried out in each biological repeat. Total number of repeats per treatment = biological repeats × technical repeats.
In order to allay the doubts of reviewers and readers, we revise the description of this sentence.
The original is “There were 9 biological repeats and 3 technical repeats in each treatment. ”, and we have revised it to “There were 9 biological repeats in each treatment, and three more technical repeats have been carried out in each biological repeat. Total number of repeats per treatment = biological repeats × technical repeats.” (in line129-131)
【Comment 3】Line 128-129: What is the basis for selecting these parameters i.e. time and temp?
[Our response 3]: Thank you for your question. The selection of these parameters is based on previous studies, and references have been added in corresponding positions.
The method have been updated this paper.
“ After the end of the treatment, the leaf, stem, and root tissues of each poplar were harvested separately, packed separately in paper bags and put into an oven for 30 min at 105°C followed by 7 days at 70°C to constant weight [9, 10]. ” (in line133-134)
【Comment 4】Line 133: Why the 5th -7th selected? Any scientific reason for the choice of the selection?
[Our response 4]: This is a very important question for us. For the sake of the rigor of the paper, we added the basis for our selection of these leaves for data measurement in the method. Without the influence of external factors, the physiological indexes of mature leaves (functional leaves) of poplar are relatively stable. In addition, the 5th-7th leaves of the poplar used in this experiment are located in the mature leaf area, so we select these leaves to determine the physiological indexes. In the previous papers[11, 12], the mature leaf areas of poplars were selected for the determination of plant resistance indexes. The selection of the 5th -7th mature leaf is based on previous studies, and references have been added in corresponding positions.
The method have been updated this paper.
“ Without the influence of external factors, the physiological indexes of mature leaves (functional leaves) of poplar are relatively stable[12]. ” (in line147-148)
Thank you for your comments. All questions are answered in the text.
-------------------------------------------------------------------------------------------------------
References
- Rahman, Z.; Singh, V. P., The relative impact of toxic heavy metals (THMs) (arsenic (As), cadmium (Cd), chromium (Cr)(VI), mercury (Hg), and lead (Pb)) on the total environment: an overview. Environ Monit Assess 2019,191, (7), 419.
- Qian, M.; Rosenqvist, E.; Prinsen, E.; Pescheck, F.; Flygare, A. M.; Kalbina, I.; Jansen, M. A. K.; Strid, A., Downsizing in plants-UV light induces pronounced morphological changes in the absence of stress. Plant Physiol 2021,187, (1), 378-395.
- Robson, T. M.; Klem, K.; Urban, O.; Jansen, M. A., Re-interpreting plant morphological responses to UV-B radiation. Plant Cell Environ 2015,38, (5), 856-66.
- He, J.; Li, H.; Ma, C.; Zhang, Y.; Polle, A.; Rennenberg, H.; Cheng, X.; Luo, Z. B., Overexpression of bacterial gamma-glutamylcysteine synthetase mediates changes in cadmium influx, allocation and detoxification in poplar. New Phytol 2015,205, (1), 240-54.
- He, F.; Zhao, Q.; Huang, J. L.; Niu, M. X.; Feng, H. C.; Shi, Y. J.; Zhao, K. J.; Cui, X. L.; Wu, X. L.; Mi, J. X.; Zhong, Y.; Liu, Q. L.; Chen, L. H.; Wan, X. Q.; Zhang, F., External application of N alleviates toxicity of Cd on poplars via starch and sucrose metabolism. Tree Physiol 2021,41, 2126-2141.
- Zhao, F. J.; Tang, Z.; Song, J. J.; Huang, X. Y.; Wang, P., Toxic metals and metalloids: Uptake, transport, detoxification, phytoremediation, and crop improvement for safer food. Mol Plant 2021.
- Ulm, R.; Nagy, F., Signalling and gene regulation in response to ultraviolet light. Curr Opin Plant Biol 2005,8, (5), 477-82.
- He, J.; Ma, C.; Ma, Y.; Li, H.; Kang, J.; Liu, T.; Polle, A.; Peng, C.; Luo, Z.-B., Cadmium tolerance in six poplar species. Environmental Science and Pollution Research 2013,20, (1), 163-174.
- Zhao, H.; Guan, J.; Liang, Q.; Zhang, X.; Hu, H.; Zhang, J., Effects of cadmium stress on growth and physiological characteristics of sassafras seedlings. Scientific Reports 2021,11, (1), 9913.
- Zhang, Z.; Yu, Z.; Zhang, Y.; Shi, Y., Impacts of Fertilization Optimization on Soil Nitrogen Cycling and Wheat Nitrogen Utilization Under Water-Saving Irrigation. Front Plant Sci 2022,13, 878424.
- He, F.; Wang, H. L.; Li, H. G.; Su, Y.; Li, S.; Yang, Y.; Feng, C. H.; Yin, W.; Xia, X., PeCHYR1, a ubiquitin E3 ligase from Populus euphratica, enhances drought tolerance via ABA-induced stomatal closure by ROS production in Populus. Plant Biotechnol J 2018,16, (8), 1514-1528.
- He, F.; Li, H. G.; Wang, J. J.; Su, Y.; Wang, H. L.; Feng, C. H.; Yang, Y.; Niu, M. X.; Liu, C.; Yin, W.; Xia, X., PeSTZ1, a C2H2-type zinc finger transcription factor from Populus euphratica, enhances freezing tolerance through modulation of ROS scavenging by directly regulating PeAPX2. Plant Biotechnol J 2019.

Reviewer 2 Report (New Reviewer)
Comments
Line 104: The modifications that authors employed different from the previous research methods should be stated to enable readers and other researchers understand such modifications in the methods.
Line 126: What do authors mean by technical repeats in each treatment?
Line 128-129: What is the basis for selecting these parameters i.e. time and temp?
Line 133: Why the 5th -7th selected? Any scientific reason for the choice of the selection?
Author Response
Responses to reviewer 2:
【Comment 1】Line 104: The modifications that authors employed different from the previous research methods should be stated to enable readers and other researchers understand such modifications in the methods.
[Our response1]: Thank you for your suggestion. In order to enable readers and other researchers to understand the modification of this method, we have supplemented the content related to the modification of the treatment method in the materials and methods.
The original is “To analyze the stimulation of UVB stress … with modification. ”, and we have revised it to “To analyze the stimulation of UVB stress on the defense mechanism of 84K poplar, and thus affect the plant's ability to absorb and tolerate Cd, we referred to previous research methods [4, 8] with modification. Due to the difference of plant species, we modified the number and frequency of UVB pretreatment and the concentration of Cd treatment. The specific treatment methods are as follows.” (in line 106-108)
【Comment 2】Line 126: What do authors mean by technical repeats in each treatment?
[Our response 2]: Thank you for your question. In order to eliminate the instrumental effects on data measurement, we have 9 biological repeats in each treatment, and three more technical repeats have been carried out in each biological repeat. Total number of repeats per treatment = biological repeats × technical repeats.
In order to allay the doubts of reviewers and readers, we revise the description of this sentence.
The original is “There were 9 biological repeats and 3 technical repeats in each treatment. ”, and we have revised it to “There were 9 biological repeats in each treatment, and three more technical repeats have been carried out in each biological repeat. Total number of repeats per treatment = biological repeats × technical repeats.” (in line129-131)
【Comment 3】Line 128-129: What is the basis for selecting these parameters i.e. time and temp?
[Our response 3]: Thank you for your question. The selection of these parameters is based on previous studies, and references have been added in corresponding positions.
The method have been updated this paper.
“ After the end of the treatment, the leaf, stem, and root tissues of each poplar were harvested separately, packed separately in paper bags and put into an oven for 30 min at 105°C followed by 7 days at 70°C to constant weight [9, 10]. ” (in line133-134)
【Comment 4】Line 133: Why the 5th -7th selected? Any scientific reason for the choice of the selection?
[Our response 4]: This is a very important question for us. For the sake of the rigor of the paper, we added the basis for our selection of these leaves for data measurement in the method. Without the influence of external factors, the physiological indexes of mature leaves (functional leaves) of poplar are relatively stable. In addition, the 5th-7th leaves of the poplar used in this experiment are located in the mature leaf area, so we select these leaves to determine the physiological indexes. In the previous papers[11, 12], the mature leaf areas of poplars were selected for the determination of plant resistance indexes. The selection of the 5th -7th mature leaf is based on previous studies, and references have been added in corresponding positions.
The method have been updated this paper.
“ Without the influence of external factors, the physiological indexes of mature leaves (functional leaves) of poplar are relatively stable[12]. ” (in line147-148)
Thank you for your comments. All questions are answered in the text.
-------------------------------------------------------------------------------------------------------
References
- Rahman, Z.; Singh, V. P., The relative impact of toxic heavy metals (THMs) (arsenic (As), cadmium (Cd), chromium (Cr)(VI), mercury (Hg), and lead (Pb)) on the total environment: an overview. Environ Monit Assess 2019,191, (7), 419.
- Qian, M.; Rosenqvist, E.; Prinsen, E.; Pescheck, F.; Flygare, A. M.; Kalbina, I.; Jansen, M. A. K.; Strid, A., Downsizing in plants-UV light induces pronounced morphological changes in the absence of stress. Plant Physiol 2021,187, (1), 378-395.
- Robson, T. M.; Klem, K.; Urban, O.; Jansen, M. A., Re-interpreting plant morphological responses to UV-B radiation. Plant Cell Environ 2015,38, (5), 856-66.
- He, J.; Li, H.; Ma, C.; Zhang, Y.; Polle, A.; Rennenberg, H.; Cheng, X.; Luo, Z. B., Overexpression of bacterial gamma-glutamylcysteine synthetase mediates changes in cadmium influx, allocation and detoxification in poplar. New Phytol 2015,205, (1), 240-54.
- He, F.; Zhao, Q.; Huang, J. L.; Niu, M. X.; Feng, H. C.; Shi, Y. J.; Zhao, K. J.; Cui, X. L.; Wu, X. L.; Mi, J. X.; Zhong, Y.; Liu, Q. L.; Chen, L. H.; Wan, X. Q.; Zhang, F., External application of N alleviates toxicity of Cd on poplars via starch and sucrose metabolism. Tree Physiol 2021,41, 2126-2141.
- Zhao, F. J.; Tang, Z.; Song, J. J.; Huang, X. Y.; Wang, P., Toxic metals and metalloids: Uptake, transport, detoxification, phytoremediation, and crop improvement for safer food. Mol Plant 2021.
- Ulm, R.; Nagy, F., Signalling and gene regulation in response to ultraviolet light. Curr Opin Plant Biol 2005,8, (5), 477-82.
- He, J.; Ma, C.; Ma, Y.; Li, H.; Kang, J.; Liu, T.; Polle, A.; Peng, C.; Luo, Z.-B., Cadmium tolerance in six poplar species. Environmental Science and Pollution Research 2013,20, (1), 163-174.
- Zhao, H.; Guan, J.; Liang, Q.; Zhang, X.; Hu, H.; Zhang, J., Effects of cadmium stress on growth and physiological characteristics of sassafras seedlings. Scientific Reports 2021,11, (1), 9913.
- Zhang, Z.; Yu, Z.; Zhang, Y.; Shi, Y., Impacts of Fertilization Optimization on Soil Nitrogen Cycling and Wheat Nitrogen Utilization Under Water-Saving Irrigation. Front Plant Sci 2022,13, 878424.
- He, F.; Wang, H. L.; Li, H. G.; Su, Y.; Li, S.; Yang, Y.; Feng, C. H.; Yin, W.; Xia, X., PeCHYR1, a ubiquitin E3 ligase from Populus euphratica, enhances drought tolerance via ABA-induced stomatal closure by ROS production in Populus. Plant Biotechnol J 2018,16, (8), 1514-1528.
- He, F.; Li, H. G.; Wang, J. J.; Su, Y.; Wang, H. L.; Feng, C. H.; Yang, Y.; Niu, M. X.; Liu, C.; Yin, W.; Xia, X., PeSTZ1, a C2H2-type zinc finger transcription factor from Populus euphratica, enhances freezing tolerance through modulation of ROS scavenging by directly regulating PeAPX2. Plant Biotechnol J 2019.

Round 2
Reviewer 1 Report (New Reviewer)
The langauge of this manuscript should be polished by native english-speakers given that there are still many places with grammatic problems.
For example,
1. L22, contens, please correct it.
2. L63-64, this meaning of this sentence is not clear, please rephrase it.
3. L71-72, the captials of the element names are wrong.
4. L122-126, the grammar of these sentences is wrong.
5. Please carefully check the language throughout the manuscript.
6. In the figures, UVB-Cd is easily misundertood into UVB without Cd, and it had better be corrected into UVB+Cd.
7. UVB increased the poplar plants to Cd toxicity, which caused great damages and affect the survial of plants. Therefore, what is the value of the Cd enrichment by UVB at the cost of damaging plant growth and development.
Author Response
Responses to reviewer 1:
【Comment 1】1. L22, contens, please correct it.
[Our response1]: Sorry, we have modified this place. And check the spelling of the word in full.
As nicely advised, the 'Introduction' section update in this paper.
The original is “The above results further…Ascorbate peroxidase (APX)).”, and we have revised it to “The above results further increased the toxicity of Cd to plants in the UVB-Cd group, which was shown as increased leaf malonaldehyde (MDA) and hydrogen peroxide (H2O2) contents, as well as down-regulated activities of antioxidant enzymes such as Superoxide Dismutase (SOD), Cata-lase (CAT), and Ascorbate peroxidase (APX)).” (in line20-24).
【Comment 2】L63-64, this meaning of this sentence is not clear, please rephrase it.
[Our response 2]: Thank you for your advice and question. We have rewritten and modified this sentence. For instance, sodium hydrosulfide (NaHS) alleviates Cd toxicity through regulations of Cd transport in Populus euphratica cells.
As nicely advised, the 'Introduction' section update in this paper.
The original is “For instance, when sodium hydrosulfide (NaHS)…significantly reduced”, and we have revised it to “For instance, sodium hydrosulfide (NaHS) alleviates Cd toxicity through regulations of Cd transport in Populus euphratica cells [1].” (in line64-65).
【Comment 3】L71-72, the captials of the element names are wrong. L71-72,
[Our response 3]: Thank you for your advice. We have changed the uppercase letters here to lowercase and checked for similar problems throughout the article.
As nicely advised, the 'Introduction' section update in this paper.
The original is “On the contrary, spraying silicon (Si) and selenium (Se) on …Cd from stems to leaves”, and we have revised it to “On the contrary, spraying silicon (Si) and selenium (Se) on leaves can effectively increase the transport ability of Cd from stems to leaves” (in line71-73).
【Comment 4】L122-126, the grammar of these sentences is wrong.
[Our response 4]: Thank you for your question. We have revised these sentences.
As nicely advised, the materials and methods update in this paper.
“The treatment models of each group in the experiment are shown in Fig.S1” (in line124-125).
“2.3. Monitoring of experimental indicators
2.3.1. Measurement of plant growth indicators
During the experiment, the height of each poplar was measured with a steel tape measure (precision 0.1cm) every 9 days. The net increase of plant height is calculated by subtracting the initial plant height from the plant height at the end of the test.” (in line 126-130).
【Comment 5】Please carefully check the language throughout the manuscript.
[Our response 5]: Thank you for your question. We have invited a native English speaker to help us polishing the paper.
【Comment 6】In the figures, UVB-Cd is easily misundertood into UVB without Cd, and it had better be corrected into UVB+Cd.
[Our response 6]: Thank you for your question.
First of all, I would like to thank the reviewers for their rigorous attitude. We were also planning to use UVB+Cd instead of UVB-Cd in the beginning. However, UVB+Cd can only represent the composite processing of UVB and Cd, but it is difficult to express the priority of their processing. This will also cause ambiguity to the reader. In addition, "-" does not mean subtraction, but means that the plant is pretreated with UVB and then treated with Cd. At the same time, we described the meanings represented by the treatment groups in Line 120-123 of the paper, and draw model diagrams for each treatment group in the supplementary material of the paper (Fig.S1) in order to increase readers' understanding of each treatment group. If the reviewer disagrees with this explanation, we are happy to let the editor make a final decision about this point.
【Comment 7】UVB increased the poplar plants to Cd toxicity, which caused great damages and affect the survial of plants. Therefore, what is the value of the Cd enrichment by UVB at the cost of damaging plant growth and development.
[Our response 7]: Thank you for your question. We have discussed this issue in the paper.
As the reviewers can see, UVB pretreatment can promote poplar to absorb more Cd, but excessive Cd must cause stress to plants. However, there was no significant difference in the dry weight of each tissue after single UVB treatment compared with CK group (Fig.S2a), indicating that proper UVB treatment was an induction signal rather than a chronic stress. Poplar stems are mostly used for wood in adulthood. However, there was no significant difference in stem dry weight between the Cd and UVB-Cd groups (Fig. S2a), indicating that UVB pretreatment did not affect the accumulation of stem biomass in poplar exposed to Cd. Hence, proper UVB pretreatment will not affect plant biomass accumulation, and make poplars absorb more cadmium ions. This result is of great significance for the remediation of heavy metals in the environment.
As nicely advised, the discussion update in this paper.
“Poplar stems are mostly used for wood in adulthood. However, there was no significant difference in stem dry weight between the Cd and UVB-Cd groups (Fig. S2a), indicating that UVB pretreatment did not affect the accumulation of stem biomass in poplar exposed to Cd. ” (Line 411-414)
Thank you for your comments. All questions are answered in the text.
This manuscript is a resubmission of an earlier submission. The following is a list of the peer review reports and author responses from that submission.
Round 1
Reviewer 1 Report
Plagiarism is considered an academic crime that is highly unethical. In this manuscript, plagiarism is detected which is about 24%. It must be less than 18% that is accepted worldwide.
Overall figures numbers are not accurately presented in the results. All figures should be below the text where they are mentioned.
However, the discussion is well stated but it can be improved with recent literature.